# Exploring the Relationships between Lifestyle Patterns and Epigenetic Biological Age Measures in Men

**DOI:** 10.3390/biomedicines12091985

**Published:** 2024-09-02

**Authors:** Te-Min Ke, Artitaya Lophatananon, Kenneth R. Muir

**Affiliations:** Division of Population Health, Health Services Research and Primary Care, School of Health Sciences, Faculty of Biology, Medicine and Health, The University of Manchester, Manchester M13 9PT, UK; te-min.ke@manchester.ac.uk (T.-M.K.); artitaya.lophatananon@manchester.ac.uk (A.L.)

**Keywords:** epigenetic biological age, biological age clock, DNA methylation, epigenetic markers, lifestyle patterns, healthy lifestyle index (HLI)

## Abstract

DNA methylation, validated as a surrogate for biological age, is a potential tool for predicting future morbidity and mortality outcomes. This study aims to explore how lifestyle patterns are associated with epigenetic changes in British men. Five biological age clocks were utilised to investigate the relationship between these epigenetic markers and lifestyle-related factors in a prospective study involving 221 participants. Spearman’s correlation test, Pearson’s correlation test, and univariate linear regression were employed for analysis. The results indicate that higher consumption of saturated fat and total daily calories, and a higher body mass index (BMI) are associated with accelerated biological aging. Conversely, higher vitamin D intake and a higher healthy lifestyle index (HLI) are linked to decelerated biological aging. These findings highlight the potential impact of specific lifestyle-related factors on biological aging and can serve as a reference for applying healthy lifestyle improvements in future disease prevention studies.

## 1. Introduction

Lifestyle patterns are associated with a wide range of chronic diseases, including cardiovascular disease, type 2 diabetes mellitus, obesity, metabolic syndrome, chronic obstructive pulmonary disease, and some types of cancer. These diseases often stem from modifiable lifestyle-related factors such as an unhealthy diet, lack of physical activity, smoking, and excessive alcohol consumption [1,2]. Therefore, it is crucial to prevent these diseases by modifying lifestyle habits. However, proactively engaging the community population to adopt healthier lifestyle habits and ensure long-term adherence to a healthy lifestyle pattern remains a challenge. Therefore, a persuasive biomarker could potentially serve as a hook to appeal to the at-risk population, encouraging participation in early disease prevention.

A reliable biomarker that can be reversed by changing lifestyle habits would not only be useful as a measurement in clinical studies but also beneficial in motivating individuals to engage in healthy lifestyle patterns. DNA methylation (DNAm), one of the epigenetic biomarkers, has been validated as a surrogate for biological age [3]. This raises questions about the potential to slow down epigenetic aging by modifying lifestyle and whether such interventions could potentially reverse the effects of aging. The current two randomised controlled trials [4,5] have demonstrated robust results with changes in biological age due to interventions. Meanwhile, another ongoing cohort study [6] used DNA methylation to discover the most effective interventions to retard biological aging. The feasibility of using DNAm as a surrogate for biological age in communication was demonstrated in previous studies [4,5,6]. 

DNAm is one of the epigenetic modifications characterised by the addition of a methyl group to the C-5 position of cytosine. This process, typically referred to as CpG methylation, occurs when a cytosine is followed by a guanine residue (CpG site) [7]. The process is catalysed by DNA methyltransferases (DNMTs), including DNMT1, DNMT3A, and DNMT3B [8]. The primary function of DNA methylation is to suppress gene expression [9]. Unlike mutations, epigenetic changes are modifiable [10]. Aberrant DNA methylation has been linked to aging and cancer [11,12]. As DNAm has emerged as a useful surrogate for “biological age”, previous researchers have proposed a number of different so-called “epigenetic clocks” to estimate age-related disease risks [13]. The epigenetic clock developed by Hannum et al. has shown a high correlation with chronological age (r = 0.96) and has been associated with all-cause mortality [14,15]. In contrast to Hannum’s blood-derived biological clock, Horvath’s first multi-tissue clock encompasses 51 different healthy tissues and cell types from 8,000 samples across 82 Illumina DNA methylation array datasets. Horvath’s first clock (Horvath 1) model also demonstrates a significant correlation with chronological age (r = 0.96–0.97) [16] and is associated with all-cause mortality [17,18]. Additionally, DNA methylation age acceleration has been found to be associated with multiple types of cancer [16]. Horvath’s second clock (Horvath 2), also known as the skin and blood clock, focuses on human skin and blood cells such as fibroblasts, keratinocytes, buccal cells, endothelial cells, blood, and saliva. This model is based on 391 CpG sites from the Illumina Infinium 450k and Illumina Infinium EPIC arrays (850K array). The skin and blood clock has also demonstrated a high correlation with age and age-related conditions [19].

Another epigenetic clock, “PhenoAge”, was invented by Levine et al. Levine’s epigenetic clock is characterised by combining clinical biomarkers with DNAm biomarkers for multi-system measurement. The PhenoAge model has shown strong associations with all-cause mortality and other aging-related outcomes such as cancer, health span, physical function, and Alzheimer’s disease. Additionally, Levine’s DNAm PhenoAge has also been strongly associated with various aging-related morbidities, including comorbidities, problems with physical functioning, the likelihood of being disease-free, and the risk of coronary heart disease [18]. 

Certain lifestyles have also been associated with DNAm PhenoAge, such as increased exercise and higher fruit/vegetable consumption, which were linked to a lower DNAm PhenoAge. DNAm PhenoAge was also found to significantly differ between smokers and non-smokers [20]. A recent biological age measurement, the DunedinPACE, was designed to measure the pace of aging. One of the features of the DunedinPACE is that the participants were of the same chronological age. The training phenotype is the 20-year pace of aging, which is composed of the longitudinal changes in 19 biomarkers, including those from cardiovascular, metabolic, renal, hepatic, pulmonary, periodontal, and immune systems at ages 26, 32, 38, and 45 years old. The DNA methylation was measured by 173 CpGs from the Illumina Infinium 450k and EPIC arrays. The DunedinPACE aims to estimate how quickly individuals are aging compared to their chronological age, which has been associated with morbidity, disability, and mortality [21]. 

Consequently, epigenetic clocks are recognised as potential tools for predicting future morbidity and mortality outcomes. It is important to note that these clocks can potentially be reversed based on lifestyle changes [4]. 

In this study, we adopted five biological age clocks, namely, the Horvath 1 [16], Hannum [14], Horvath 2 [19], PhenoAge [20], and DunedinPACE [21], as measures of DNAm to investigate the relationship between different epigenetic biological clocks and lifestyle-related factors. The aim of this study was to explore the correlation between epigenetic biomarkers and lifestyle-related habits in British men, with the intention of evaluating suitable epigenetic biomarkers as measures of healthy lifestyle patterns in our second-phase cancer prevention study. 

## 2. Materials and Methods

### 2.1. Study Population

A total of 221 male participants were included in the study. The recruitment process was conducted in collaboration with the Graham Fulford Charitable Trust (GFCT). To recruit participants for this study, the GFCT advertised the events on their website (https://gfct.mypsatests.org.uk/Events/ accessed on 1 March 2023). Prior to attending the events, participants were informed about our study. Research representatives were available at the screening events to provide additional explanations as needed. The eligibility criteria included being males, being aged 40–79 years, having the ability to read and write English to understand all the provided written information, and being able to complete our questionnaires and provide a signed consent form. The exclusion criteria included a history of any type of cancer and/or psychiatric disease. The exclusion of individuals with a psychiatric history aimed to prevent any distress that may have arisen during this study. 

All the participants in this study provided consent. The participants were asked to complete a screening questionnaire at the recruitment site. Blood samples were collected. To evaluate factors related to lifestyle, the food frequency questionnaire (FFQ) was used. The FFQ was adopted from the European Prospective Investigation into Cancer and Nutrition (EPIC) FFQ [22]. A Lifestyle Pattern Questionnaire was also used. The questionnaire was modified from the Health Behaviour and Stages of Change Questionnaire (HBSCQ) [23]. The FFQ and Lifestyle Pattern Questionnaire were sent to the participants by email via a link to Research Electronic Data Capture (REDCap) [24] on the University of Manchester website at https://redcap.manchester.ac.uk/ (accessed on 1 March 2023). Figure 1 illustrates the subject recruitment and study process. The entire process of participant recruitment and an introduction to the UNIMAN pilot study can be found in our YouTube introduction at https://youtu.be/ci4-Wi-fNkc (accessed on 1 March 2023).

### 2.2. Assessment of Lifestyle-Related Exposures 

The lifestyle-related exposures evaluated in the epigenetic biomarker included body mass index (BMI), tobacco smoking, alcohol consumption, physical activity, dietary intake of cholesterol, saturated fat, sugar, vitamin D, and total calories, and the healthy lifestyle index (HLI). 

BMI data were obtained from the screening questionnaire. Tobacco smoking status, alcohol consumption, and physical activity were obtained from the Lifestyle Pattern Questionnaire. The dietary intake of cholesterol, saturated fat, sugar, Vitamin D (VitD), and total daily calories were derived from the FFQ. The calculation of nutrient intake from the food frequency questionnaire, based on average portion sizes and frequency of consumption to derive the daily average of intake for each reported nutrient, was conducted. In the analysis of the correlation between each lifestyle-related exposure and biological aging, tobacco smoking status, alcohol consumption, and physical activity were used as categorical variables. The details of the classification are described in Appendix A. BMI and intake of cholesterol, saturated fat, sugar, Vitamin D (VitD), and total daily calories were used as continuous variables.

In this study, how the composition of lifestyle patterns affects biological aging was also explored. Therefore, the healthy lifestyle index (HLI) established in this study was based on each lifestyle-related exposure described above. The HLI was established by combining all the categorical lifestyle-related exposures with their respective weights. The details of the classification for each lifestyle-related exposure are described in Appendix A. The weighted score was designed so that a healthier pattern corresponds to a higher score, as listed in Appendix A. The total scores ranged from 10 to 22. High HLI scores indicate adherence to healthier lifestyle patterns compared to low HLI scores.

### 2.3. Blood Sample Analysis: Epigenetic Data Process and Quality Control

Blood samples were processed to extract DNA. The Infinium Methylation EPIC v2.0 Kit (935,000 CpG array) [25] was used to obtain DNA methylation data. The meffil R Package [26,27], designed for analysing DNA methylation data from Infinium 450, EPIC v1, and v2 BeadChips, was used for quality control check and generated beta values from methylation data [27]. The procedures included background and dye bias correction, sex prediction, cell count estimates, sample quality checks, and normalisation. Normalisation involved determining the number of principal components to include in the quantile normalisation and then removing control probe variance from the sample quantiles. Additionally, problematic CpGs identified during the QC analysis were eliminated [26].

Sample quality checks included comparing median methylation and unmethylated intensity. The outlier threshold was set at more than 3 standard deviations (SD) from the expected median methylated signal. Control probes were grouped into 42 categories, with outliers defined as deviations beyond the mean ± 5 SD. To determine the proportion of probes that did not pass the detection *p*-value for each sample, a detection *p*-value threshold was set at 0.01. Additionally, to identify the proportion of probes not meeting the bead count threshold, this was defined as a bead number lower than 3. Regarding probe quality checks, the proportion of samples with a detection *p*-value greater than 0.01 was calculated. Similarly, the proportion of samples with a low number of beads, set as fewer than 3, was also determined [27]. In this study, no sample was removed, and high-quality methylation data were obtained. 

The biological clocks using the Horvath 1 [16], Hannum [14], Horvath 2 [19] and PhenoAge [20] methods were calculated using the methylCIPHER R package [28]. The biological clock using the DunedinPACE [21] methods was computed by using the DunedinPACE R package [29]. The average present CpG rates were 96%, 90%, 96%, 96%, and 84% for Horvath 1, Hannum, Horvath 2, PhenoAge, and DunedinPACE, respectively.

### 2.4. Data Analysis and Statistics

All DNA methylation processes and biological clock calculations were completed by R software [30] (R version 4.2.1, R Development Core Team, Vienna, Austria). All lifestyle-related variables and biological clock ages were initially examined for their distribution pattern using the Shapiro–Wilk test, as well as Skewness and Kurtosis tests [31]. 

In this study, the epigenetic age acceleration (AA) for each clock was calculated as the difference in age (Δ Age) between the biological age (BA) and chronological age (CA) (Δ Age = BA – CA) [32,33]. A positive AA indicates that biological aging occurs faster than chronological aging, while a negative AA suggests that biological aging is slower than chronological aging. Additionally, comparisons between biological age and chronological age were also measured in two states: BA greater than CA, and BA less than or equal to CA. The abbreviation “Diff” is used to denote these differences.

In the correlation analysis, categorical variables and those with a non-normal distribution were analysed using Spearman’s correlation test, while continuous variables with a normal distribution were analysed using Pearson’s correlation test [34]. Univariate linear regression was conducted on variables that showed significant correlation in the Pearson test. In this study, the R packages “corrplot”, “ggplot2”, and “pheatmap” were employed to generate the heatmap matrix.

A *p*-value less than 0.05 was considered “statistically significant”, and 95% confidence intervals (95% CI), not including one, were also used to guide the delineation of statistical significance. However, for the correlation analysis between the variables and accelerated aging (AA) across different biological clocks, the false discovery rate (FDR) adjusted *p*-value (*q*-value) was used to reduce the risk of Type I errors due to multiple testing and to enhance the robustness of the findings [35,36]. All statistical analyses were performed using STATA/MP software [37] version 17 (StataCorp LLC, College Station, TX, USA.) and R software [30] (R version 4.2.1, R Development Core Team, Vienna, Austria). 

### 2.5. Ethics

This study was granted approval from the University of Manchester Research Ethics Committee 3 on 4 May 2022, and two subsequent amendments were approved on 14 February 2023 and 8 June 2023. The ethics approval reference number is 2022-13644-23283.

## 3. Results

### 3.1. Participants in the Study 

Details of the participants in this study are presented in Table 1. Initially, a total of 222 male participants were recruited across six sessions. However, one participant withdrew following the recruitment, resulting in a total of 221 participants.

Regarding the age distribution in Table 1, the mean age in this study was 61.6 years old, with a range from 43 to 79. The majority of the participants were in the 50–59 and 60–69 age groups, representing 34.84% (77 participants) and 35.75% (79 participants) of the total, respectively. The 70–79 age group comprised 21.72% (48 participants), while the 40–49 age group was the smallest, at 7.69% (17 participants). In terms of gender, the study included only male participants since it was advertised through the original PSA test programme.

A high level of consent was observed for joining the Lifestyle Pattern and Epigenetic Biomarker Study group, with 214 participants (96.83%) consenting. However, participation in the questionnaires demonstrated a relatively lower engagement rate, with the Food Frequency Questionnaire completion rate at 69.68% (154 participants) and the Lifestyle Pattern Questionnaire completion rate at 66.51% (147 participants) (Table 1).

### 3.2. The Distribution of Lifestyle-Related Factors Analysed in Relation to Epigenetic Biological Age

Table 2 presents the distribution of various lifestyle-related factors among the study participants. Regarding BMI, among the 214 participants who provided weight and height information, 59 (27.57%) were classified as obese and 94 (43.93%) as overweight, and the remaining 61 (28.5%) fell within the normal range. The mean BMI among all the participants was 28.30, which was classified as overweight. In terms of smoking status, among the 147 participants who reported their smoking habits, more than half had never smoked (n = 91, 61.90%) and 36.05% (n = 53) were former smokers. However, only about 2% (n = 3) were identified as current smokers. Alcohol consumption patterns were observed in 139 participants who responded to questions related to drinking habits. Of these, approximately half (n = 76) consumed no more than seven drinks per week, while 41.01% (n = 57) reported consuming more than seven drinks per week. Additionally, about one-twentieth (n = 6) reported never having alcohol-drinking habits. Among the 139 participants who reported their physical activity levels, the majority (79.14%, n = 110) exercised routinely (at least three times a week for 30 min a day). Approximately one-fifth (n = 27) engaged in exercise occasionally but not regularly, while only a minority of 1.44% (n = 2) did not engage in any exercise at all.

Regarding the dietary habits of the 154 participants who completed the FFQ, with respect to cholesterol levels (Table 2), around 80% (n = 122) of the participants had cholesterol intakes below or equal to the recommended level of 300 mg/day and the remaining one-fifth (n = 32) had levels above 300 mg/day. The average daily cholesterol intake was around 227 mg, which meets the previous National Institute for Health and Care Excellence (NICE) recommendation of staying under 300 mg/day [38]. Saturated fat intake (Table 2) indicated that 26.62% of the participants (n = 41) consumed more saturated fat than recommended by the UK National Health Service (NHS) guidelines, with males exceeding 30 grams (g) per day (d) [39]. In contrast, the majority of the participants (n = 113, 73.38%) consumed less than these thresholds. The average saturated fat consumption was 26.05 g/day. In terms of total sugar intake (Table 2), approximately one-third of the participants (n = 48) consumed less than 90 g/d, aligning with NHS recommendations [40]. However, more than two-thirds of the participants (n = 106) exceeded this amount. The average total sugar consumption was 116.74 g.

Regarding Vitamin D intake levels (Table 2), 87.01% of the participants (n = 134) consumed less than 10 micrograms (mcg), while only approximately 13% (n = 20) achieved intake levels higher than 10 mcg, as advised by the NHS [41]. The average Vitamin D intake in this study was 4.98 mcg per day. Regarding total daily caloric intake, three-quarters of the participants (n = 117) met the NHS recommendation of no more than 2500 kcal for males and 2000 kcal for females [42]. Nonetheless, a quarter of the participants (n = 37) consumed more than the NHS recommendation. The average caloric intake in this study was 2129.79 kcal/day. Among the 133 participants who reported all the elements required to calculate the healthy lifestyle index (HLI) in this study, the average HLI score was 18.29, with scores ranging from 14 to 22.
biomedicines-12-01985-t002_Table 2Table 2The distribution of lifestyle-related factors.Lifestyle-Related FactorsNumber of Participants**Body mass index (BMI) (n = 214)**
Obese (BMI ≥ 30)59 (27.57%)Overweight (25 ≤ BMI < 30)94 (43.93%)Normal (18.5 ≤ BMI < 25)61 (28.50%)**BMI (con) (n = 214)**28.30 ± 5.78 (14.98–61.31) ^#^**Smoking status (n = 147)**
Current smokers3 (2.04%)Previous smokers53 (36.05%)Never91 (61.90%)**Alcohol consumption (n = 139)**
More than 7 drinks/week ^†^57 (41.01%)No more than 7 drinks/week ^†^76 (54.68%)Never6 (4.32%)**Physical activity (n = 139)**
No exercise habit2 (1.44%)Exercise occasionally27 (19.42%)Exercise routinely (at least 3 times a week, 30 min a day)110 (79.14%)**Cholesterol (n = 154)**
>300 mg/day32 (20.78%)≤300 mg/day122 (79.22%)**Cholesterol (con) (n = 154)**227.46 ± 113.49 (59.91–809.13) ^#^**Saturated fat (n = 154)**
Male > 30 g/day 41 (26.62%)Male ≤ 30 g/day 113 (73.38%) **Saturated fat (as continuous) (n = 154)**26.05 ± 10.97 (9.3–79.68) ^#^**Total sugar (n = 154)**
>90 g/day106 (68.83%)≤90 g/day48 (31.17%)**Total sugar (as continuous) (n = 154)**116.74 ± 42.21 (42.07–302.34) ^#^**Vitamin D (n = 154)**
<10 mcg/day134 (87.01%)≥10 mcg/day20 (12.99%)**Vitamin D (as continuous) (n = 154)**4.98 ± 4.07 (0.66–23.28) ^#^**Daily total calories (n = 154)**
Male > 2500 kcal/day, Female > 2000 kcal/day37 (24.03%)Male ≤ 2500 kcal/day, Female ≤ 2000 kcal/day117 (75.97%)**Daily calories (as continuous) (n = 154)**2129.79 ± 609.20 (1020.43–4194.43) ^#^**Healthy life index (HLI) (as continuous score) (n = 133)**18.29 ± 1.72 (14–22) ^#^^†^ One unit equals 10 ml or 8 g of pure alcohol [43]; ^#^ Mean ± standard error (range).


### 3.3. The Results of Epigenetic Markers 

Table 3 presents the descriptive statistics for the five biological clocks, their associated accelerated aging (AA), and the Spearman’s correlation between biological age and chronological age. These clocks exhibited diversity in average means and spanned various ranges, each with distinct minimum and maximum ages. The Horvath 1 clock had the highest mean value of 64.22 ± 8.18 years, while the Hannum clock presented the lowest mean value of 34.22 ± 9.69 years. The Horvath 1 clock recorded the highest maximum age at 84.04, whereas the PhenoAge clock revealed a notably low minimum age of 27.02. The DunedinPACE, with a mean close to 1, indicated the pace of aging, placing the participants from 0.76 fold per year to around 1.4 fold per year. 

In terms of correlation to chronological age, since both the biological clock age and chronological age were not normally distributed according to the Shapiro–Wilk test and Skewness and Kurtosis tests, Spearman’s correlation test was therefore reported. The Horvath 2 clock displayed the strongest correlation with age (r = 0.909, *p* < 0.05), followed by the Horvath 1 clock (r = 0.860, *p* < 0.05), the PhenoAge clock (r = 0.802, *p* < 0.05), and the Hannum clock (r = 0.693, *p* < 0.05) (Table 3). 

Age acceleration (AA) was calculated for each biological clock except the DunedinPACE, as this was originally designed to represent the pace of aging. AA was defined as the difference between the epigenetic biological age (BA) and chronological age (CA). Therefore, negative values indicated a slower rate of biological aging relative to chronological age, and vice versa. In this study, the mean AA for the Hannum clock, Horvath 2, and PhenoAge were all negative (Table 3), indicating a slower biological aging process. Moreover, the range of the Hannum AA spanned −42.03 to −0.37 (Table 3), suggesting that the biological age of all the participants was lower than their chronological age.
biomedicines-12-01985-t003_Table 3Table 3The descriptive measures for different biological clocks and accelerated aging.
MeanSDMinimumMaximumCorrelation ^#^ with Chronological Age**Biological Clocks**




Horvath 1 [16]64.228.1836.3084.040.860 *Hannum [14]34.229.6916.3174.630.693 *Horvath 2 (Blood & Skin Cock) [19]51.407.9034.6674.630.909 *PhenoAge [20]54.839.6427.0282.160.802 *DunedinPACE [21]1.010.100.761.390.205 ***Accelerated aging (AA) ^†^**




Horvath 1 AA2.824.48−8.7516.68-Hannum AA−27.197.68−42.03−0.37-Horvath 2 AA−10.003.61−24.070.76-PhenoAge AA−6.575.95−22.0310.04-DunedinPACE AA-----Notes: SD = standard deviation. ^#^ Correlation was measured using Spearman’s correlation test. ^†^ Accelerated aging (AA) is defined as the difference between biological clock age and chronological age. * *p*-value < 0.05.


Table 4 and Figure 2 presents a comparison between the five biological clock ages and chronological age. The Hannum, Horvath 2, and PhenoAge clocks predominantly indicated slower biological aging, whereas the Horvath 1 and DunedinPACE suggested that the majority are aging faster biologically (Table 4 and Figure 2). Specifically, the Hannum clock uniquely showed that 100% of the participants had a biological age less than or equal to their chronological age. Similarly, the Horvath 2 clock revealed that the vast majority (99.53%) of the participants had a biological age that was younger than their chronological age. The PhenoAge data indicated that approximately 87.4% of the participants (n = 187) were biologically younger than or the same age as their chronological age.

Conversely, for the Horvath 1 clock, a dominant proportion of the participants, about 74% (n = 158), exhibited a biological age greater than their chronological age (Table 4 and Figure 2). In addition, more than half of the participants (n = 116, 54.21%) showed accelerated biological aging according to the DunedinPACE clock, as shown in Table 4 and Figure 2.
biomedicines-12-01985-t004_Table 4Table 4Comparison of different biological clock ages with chronological ages.
Biological Clock Age Greater than Chronological AgeBiological Clock Age Less than or Equal to Chronological AgeTotalNumber**Horvath 1 [16]**158 (73.83%)56 (26.17%)214**Hannum [14]**0 (0%)214 (100%)214**Horvath 2 [19]**1 (0.47%)213 (99.53%)214**PhenoAge [20]**27 (12.62%)187 (87.38%)214**DunedinPACE [21]**116 (54.21%)98 (45.79%)214


The heatmap matrix in Figure 3 displays the Spearman’s correlation coefficients for various biological clock ages and biological clock aging acceleration (AAs). Figure 3A illustrates the positive correlations among the four biological clocks, all of which are statistically significant. Figure 3B reveals that all the clock AAs are significantly positively correlated with each other (all *p*-values < 0.05), although the relationships are comparatively weaker than those between the biological clock ages in Figure 3A. The strongest positive correlation was observed between the Horvath 2 AA and Hannum AA (r = 0.56, *p*-value < 0.05), while the weakest was between the PhenoAge AA and Hannum AA (r = 0.14, *p*-value < 0.05). The DunedinPACE was not included in this heatmap matrix because it measures the pace of aging, focusing on the rate of biological aging. This concept differs from the biological aging measures represented by the Horvath 1, Hannum, Horvath 2, and PhenoAge.

### 3.4. Correlation between Various Variables and Accelerated Aging (AA) for Different Biological Clocks

Table 5 demonstrates the Spearman’s correlation coefficients between various variables and accelerated aging (AA) for different biological clocks. For the Horvath 1 AA, significant weak positive correlations were observed with the continuous variables of saturated fat (r = 0.241, *p*-value < 0.05) and total daily calories (r = 0.221, *p*-value < 0.05). These results suggest that higher levels of saturated fat and total calorie intake are correlated with accelerated biological aging, as measured by the Horvath 1 clock. 

In the Horvath 2 AA, weak positive correlations were found with the continuous variable of saturated fat (r = 0.199, *p*-value < 0.05), similar to the results from the Horvath 1 AA. These findings suggested that higher levels of saturated fat may be associated with faster biological aging rates as indicated by the Horvath 2 clock. However, a weak negative correlation was found between smoking and the Hannum AA (r = −0.234, *p* < 0.05).

Regarding the DunedinPACE AA in Table 5, it showed weak positive correlations with BMI in the continuous variable of BMI (r = 0.371, *p*-value < 0.05), indicating a relationship between higher BMI and faster biological aging in the DunedinPACE model. Additionally, a weak negative correlation was observed between the continuous variable of Vitamin D (r = −0.199, *p*-value < 0.05) and the DunedinPACE AA, supporting the association of higher Vitamin D levels with a slower pace of biological aging in the DunedinPACE model.

Overall, higher saturated fat was related to faster biological aging in the Horvath 1. Increased total daily calorie intake was associated with accelerated biological aging in the Horvath 1 clocks. A higher BMI was observed to correlate with faster aging in the DunedinPACE clock. On the other hand, higher Vitamin D consumption was associated with slower biological aging in the DunedinPACE model.

Furthermore, we computed all the variables into the healthy lifestyle index (HLI) for this study. The HLI was examined as normal distribution using the Shapiro–Wilk test (*p*-value > 0.05) and Skewness and Kurtosis tests (*p*-value > 0.05) first. Consequently, a Pearson correlation test was further performed between the different biological AAs and the HLI. The results in Table 6 demonstrate weak negative correlations between the HLI and the PhenoAge (r = −0.278, *p* < 0.01) and DunedinPACE (r = −0.261, *p* < 0.05) results. Hence, the findings suggested that a higher HLI, indicative of healthier lifestyle habits, is associated with slower biological aging. 

Moreover, the significant results presented in Table 7 were further analysed using univariate linear regression to examine the relationship between the HLI and the PhenoAge AA and DunedinPACE methods. The univariate linear regression result is shown in Table 7. The beta coefficients suggest that there is a significant inverse relationship between the HLI and both the PhenoAge clock and DunedinPACE model. Specifically, a one-unit increase in HLI was associated with a decrease in the PhenoAge AA by 0.780 (*p*-value = 0.001) and a decrease in the DunedinPACE results by 0.01 (*p*-value = 0.039). These findings imply that adherence to a healthier lifestyle, as indicated by a higher HLI score, may contribute to a slower rate of biological aging, according to these models.

## 4. Discussion

The purpose of this study was to explore the relationship between lifestyle patterns and epigenetic biological age in British men. In the analysis of lifestyle-related factors, the selected nutrient components showed a correlation with biological age. Higher consumption of saturated fat and total daily calories were both related to faster biological aging. Elevated vitamin D intake was correlated with decelerated biological aging. Additionally, an increase in biological aging was observed with a higher BMI. Combining all the lifestyle-related variables into the healthy lifestyle index (HLI), a higher HLI indicates a better lifestyle pattern, which is associated with decelerated biological aging.

The high consent rates for joining the Lifestyle Pattern and Epigenetic Biomarker Study group (96.8%) suggest a strong interest in epigenetic biomarker research. This interest might stem from participants initially joining for the PSA test, thus showing a pre-existing curiosity about biomarker tests. Additionally, our research team have delivered several speeches at GFCT public events, including discussions on genetic cancer risk and the role of epigenetic biomarkers in indicating biological age, potentially influencing lifestyle habits. These efforts could have motivated participation in the study. Further details on these talks are available on the YouTube channel (https://youtu.be/XUqql1gRB2A, accessed on 1 March 2023), which may also have incentivised individuals to join the study. However, lower completion rates for the FFQ (~70%) and Lifestyle Pattern Questionnaire (~67%) were observed. These questionnaires, sent via email link, might not be preferred by the senior population, who may be less inclined to use computers for filling in questionnaires. Additionally, with the FFQ comprising 17 pages, it is possible that some participants who completed the FFQ did not proceed to the Lifestyle Pattern Questionnaire.

The distribution of the lifestyle-related factors in Table 2 highlights various lifestyle habits and health metrics of the participants. Due to some participants opting not to answer, the total numbers vary across variables. Notably, around 62% of the participants declared that they had never smoked, with only three participants identified as current smokers. This contrasts with the UK general population’s lifestyle patterns, where about 12.9% of adults were current smokers in 2022 [44]. Around 59% of the participants drink no more than seven drinks a week or have no drinking habits. According to the National Health Service (NHS) guidelines (in which one drink equals 1.9 units based on the average size of one can of beer, one bottle of beer, and one glass of wine) [44], seven drinks equate to approximately 13.3 units (7 drinks × 1.9 units = 13.3 units). This indicates that more than half of our participants (59%) follow the NHS’s advice of consuming no more than 14 units a week [44]. Additionally, about 99% of the participants reported that they exercised, with around 79% exercising regularly (at least three times a week, 30 min a day). In addition, according to the 2021 NHS England health survey [45], 57% of adults drank at a low-risk level (more than 14 units per week). According to a 2022 survey from SPORT ENGLAND [46], 25.8% of the population is categorised as inactive (less than 30 min of exercise a week). Therefore, our study population evidently has relatively healthy lifestyle habits. Even so, about 72% of the participants remain categorised as overweight or obese.

Regarding dietary habits, among the 154 participants who completed the FFQ, as shown in Table 1, 21% and 27% exceeded the NHS advice on cholesterol and saturated fat intake, respectively. Additionally, around one-quarter of the participants consumed more daily calories than recommended by the NHS [39,47]. Notably, 69% ingested more than 90 g of total sugar per day, surpassing the NHS’s maximum suggested threshold [40]. Furthermore, 87% of the participants did not meet the recommended intake of 10 mcg of Vitamin D per day [41]. Overall, the majority of the participants consumed too much sugar and insufficient amounts of Vitamin D. At least one-quarter of the participants consume excessive calories, saturated fat, or cholesterol daily, indicating a need for dietary improvements among most participants.

The analysis in Table 3 showcases the diversity in means and ranges among the five biological clocks, which aligns with the results of a previous study [48]. The Horvath 2 clock exhibits the strongest correlation (r = 0.91) with chronological age, a similarity echoed in a prior Health and Retirement cohort study [48]. The high correlation with chronological age in the Horvath 1 (r = 0.86) and 2 (r = 0.91) and Hannum (r = 0.7) clocks may stem from these models being trained using chronological age [16,19]. The PhenoAge model, trained with mortality data, also showed a strong relationship (r = 0.8) with chronological age [20]. A relatively weaker correlation (r = 0.2) with chronological age presented in the DunedinPACE clock may be attributed to its unique approach to measuring the pace of aging [21]. In summary, all five biological clock ages were positively related to chronological age, with the Horvath 1 and 2, Hannum, and PhenoAge exhibiting stronger relationships.

The epigenetic age acceleration (AA) for each clock in this study was determined by calculating the difference between biological and chronological ages [32]. Thus, a positive AA indicates faster biological aging, and vice versa. The DunedinPACE clock was designed to quantify the pace of biological aging, which is interpretable as a rate of 1 year of biological aging per year of chronological aging. Values greater than one indicate accelerated aging, and vice versa. Consequently, there is no need to calculate the difference between biological and chronological ages for the DunedinPACE clock in this study. 

The heatmap matrix in Figure 3 shows the positive relationships between each biological clock. The DunedinPACE clock’s unique approach to measuring the pace of aging, therefore, is not discussed in the heatmap matrix. The phenomenon aligns with the findings from a previous Health and Retirement study [48]. However, generally weak correlations in accelerated aging (AA) in Figure 3B were found. This may suggest a divergence in the direction of aging across the clocks. The consistent discrepancy between the biological clocks and chronological age is illustrated in Table 4 and Figure 2, revealing contrasting aging patterns among the 214 participants. Despite the fact that the Hannum, Horvath 2, and PhenoAge clocks generally suggest slower biological aging, the Horvath 1 and DunedinPACE indicate a tendency towards faster biological aging. Notably, the Hannum clock implies all the participants are aging at or below their chronological rate, in contrast to the Horvath 1 and DunedinPACE, where the majority of the participants exhibit accelerated aging. The diversity may be caused by the variety in methodology design, tissue use, CpG site selection, and training aim outcome in each biological clock [49]. In addition, in our study, the DNA methylation data were obtained using the Infinium Methylation EPIC v2.0 Kit, which covers 935,000 CpG sites. The methylation array used in our study differs from those used in the Horvath 1, Hannum, Horvath 2, PhenoAge, and DunedinPACE. As a result, some of the CpG sites present in these five biological clocks are not detected in our study. Although the rate of missing CpG sites is not high, it can still affect the results if the missing CpGs have a significant impact on the clocks.

The results in Table 5 intricately link lifestyle habits with various biological clock aging metrics. However, the findings on nutritional components underscore the potential impact of diet habits on the biological aging process. Increased saturated fat consumption is related to faster aging in both the Horvath 1 and Horvath 2 clocks. Higher daily calorie intake is related to faster biological aging according to the Horvath 1 clock. Additionally, adequate Vitamin D intake is observed to correlate with decelerated aging in the DunedinPACE model. The findings are consistent with a previous study exploring the relationship between healthy eating patterns and epigenetic biological age [50]. The researchers discovered that adherence to all four healthy diets—DASH [51], Healthy Eating Index–2015 [52], Alternative Healthy Eating Index (aHEI-2010) [53], and the Alternative Mediterranean Diet [54]—was linked with slower epigenetic age acceleration, as measured by the PhenoAge and GrimAge [55] clocks. In addition, a study [56] on a Women’s Health Initiative identified several associations between specific foods and aging. Organ meat, egg, sausages, cheese, and lunch meat were associated with accelerated aging, while peaches/nectarines/plums and nuts were associated with decelerated aging. Each of the above studies has unique features but shares key principles of healthy eating [57]. All emphasise a low intake of saturated fats, moderate calorie consumption, and the importance of nutrient-rich foods, including those high in Vitamin D. Generally, these diets advocate for a rich intake of fruits, vegetables, whole grains, and lean proteins, in line with recommendations to reduce saturated fat levels.

Additionally, a randomised controlled caloric restriction study [58] involving 220 adults, with an intervention of 25% calorie restriction for 2 years, demonstrated a slower DunedinPACE aging rate. Another study [59] randomly assigned 278 healthy individuals to an elderly-tailored Mediterranean Diet intervention for one year, resulting in a 0.58-year decrease in their customised epigenetic biological age. Therefore, combining our results with previous studies’ findings, we can see that the healthiness of a diet can decrease biological aging.

In this study, we focus on specific nutrient components, including saturated fat, cholesterol, sugar, vitamin D, and total calories, rather than all possible foods or nutrients. These nutrients have been shown to significantly affect health outcomes, particularly those related to aging and chronic diseases [60,61]. This targeted approach allows for a more in-depth understanding of the relationships between these key nutrients and biological aging. On the other hand, we use non-energy-adjusted intakes instead of energy-adjusted intakes because this study focuses on absolute nutrient intake or dietary patterns without controlling for total energy intake. We aim to retain the information about the actual amount of nutrient intake, which is important for understanding the direct relationship with the reported intake levels. Additionally, since the demographic characteristics of our study population do not vary significantly, using absolute nutrient intake is relevant. As the FFQ was administered by REDCap, which allowed us to specify the data input completion prior to submission, there were no missing FFQ data in our study. In future studies, a blood-based biomarker of these nutrients could be added to provide a clear association with epigenetic clocks. 

In this study, physical activity habits did not show a significant correlation with biological clock ages. This may have reflected the fact that the cohort was recruited based on attendance at a PSA test event, indicating some degree of health consciousness among the enrolled participants, which may be one of the reasons why only 2% of the participants did not engage in exercise habits. Furthermore, in this study, self-reported questionnaires were utilised, and the classification of physical activity levels could not be evaluated based on age-specific recommendations due to the design of the questions. These factors could potentially result in a higher level and lack of variation in exercise levels within the cohort, thus impacting the ability to detect correlations. Aligned with our results, a previous twin study [62] showed that epigenetic aging did not significantly impact aging deceleration through leisure time physical activity as measured by the Horvath 1 clock. Additionally, another study [63] also found no cross-sectional correlations between physical activity markers and Horvath 1 biological aging. Nevertheless, one study [64] supported that a six-month exercise intervention led to changes in DNA methylation (DNAm) in human adipose tissue. Another study [65] demonstrated that lifelong regular physical activity is associated with DNA methylation changes in human skeletal muscle tissue. Although epigenetic aging clocks have not yet demonstrated a relationship with physical activity in previous studies [62,63] and in our results, some studies [64,65] have still suggested the potential benefits of DNA methylation changes led by exercise [57]. 

Despite no correlation between exercise and epigenetic aging in our findings, epigenetic accelerated aging showed a significant positive association with BMI, as measured by the DunedinPACE. Consistent with this, the Health and Retirement study [48] also observed a faster biological aging pace in overweight and obese individuals. Furthermore, increases in epigenetic age acceleration with a higher BMI were reported in the liver [66] and visceral adipose tissue [67] in the Horvath 1 clock.

Smoking has been shown to be related to epigenetic signature changes in various studies [68,69,70,71]. Epigenetic aging was also observed to be linked to lifetime exposure to smoking [72]. Alcohol reduction is correlated with decelerated biological aging [57]. A previous study [73] addressed that cumulative liquor and total alcohol consumption are associated with epigenetic age acceleration. However, smoking and alcohol consumption did not show a significant positive relation to biological clock age in this study. The fact that only 2% of the participants were current smokers meant that we were not able to investigate this exposure. For alcohol, the lack of classification of heavy drinkers in a separate category may also need to be addressed.

Integrating all lifestyle-habit-related exposure variables into an HLI in this study was used as an overall indicator of a healthy lifestyle pattern. The HLI has been used in various studies [74,75,76,77,78] to assess the impact of lifestyle choices on health outcomes, which is often applied in risk assessment [74,75], comparing the health impacts of different HLI [77,78] and longitudinal monitoring [76]. A higher HLI indicates a tendency towards a healthier lifestyle pattern, which appeared to be associated with decelerated biological aging in the PhenoAge and DunedinPACE. A previous randomised controlled study (RCT) [4] involving 43 adult males, with interventions including diet control, exercise, and stress management for 8 weeks, demonstrated a reversal in Horvath 1 biological age by approximately 3 years. Additionally, a case series [79] revealed that dietary and lifestyle interventions for 8 weeks in six individuals showed an average decrease of 4.60 years in the Horvath 1 clock. Therefore, the findings of our study and previous intervention studies [4,79] both suggest that adopting a healthy lifestyle may potentially lower biological age. 

Other biological aging measures that use clinical biomarkers instead of DNA methylation also show the relationship between lifestyle patterns and biological aging. A study [80] using data from the Health and Retirement study found that lifestyle behaviours had the most significant contribution to the variance in the PhenoAge outcomes. Another study [81] using a multidimensional approach also highlighted the impact of lifestyle-related features on biological aging processes.

As the HLI can provide a general overview of an individual’s lifestyle pattern, and it has been proven that a higher HLI is associated with slower biological aging in two different biological aging clocks (PhenoAge and DunedinPACE in this study), the HLI and the biological aging measurements of the PhenoAge and DunedinPACE will be considered for adoption in the second phase of this study to compare the changes in biological aging before and after joining the Cancer Prevention Study group. Furthermore, the epigenetic mutation load (EML), also known as the total number of stochastic epigenetic mutations (SEMs), has also been used as an epigenetic biomarker [5]. Nevertheless, the relationship between lifestyle-related habits and EML was not investigated in this study as the study aimed to focus on the biological aging clock.

The strengths of this study include its inherent design and the comprehensive collection of lifestyle-related factors. Additionally, the quality control of epigenomic data consistently demonstrates high data quality, making the sequential analysis results of the DNAm reliable. Moreover, since different epigenetic aging clocks were trained with various outcome aims and on different CpG sites, this study adopted various epigenetic aging clock methods to investigate the correlation with lifestyle-related variables. Furthermore, the participants showed a high response rate to attending the Lifestyle Pattern and Epigenetic Biomarker Study, achieving the target sample sizes. Finally, to our knowledge, this is the first study to explore the relationship between biological aging, dietary nutrient components, and the healthy lifestyle index.

There were some limitations to this study. First, this study includes only males because recruitment was through a community-based PSA testing programme for men. Therefore, to generalize the conclusions to females, it is necessary to recruit females for analysis, which is part of our planned further work. Second, even though consent from the participants for the Lifestyle Pattern and Epigenetic Biomarker Study group was high, the response rates for the FFQ and Lifestyle Pattern Questionnaires were relatively lower, despite reminders. This may cause response bias and selection bias in our results analysis. Therefore, in our future recruitments, there is a need to encourage participants to complete the questionnaires. For those who have problems using a computer, offering assistance or posting them a paper-based questionnaire to increase the completion rate needs to be considered. Third, since all the lifestyle-related information was collected from self-reported data, this may lead to recall and reporting biases. Fourth, even though the presence rate of CpG in each biological clock was generally high, a relatively low presence rate (84%) in the DunedinPACE was noted. Considering that this may influence the results, a computation method will be adopted in subsequent studies. Fifth, as we did not use a multivariable-adjusted method, some potential confounders may still exist in this study and should be considered in future research. Nevertheless, this study is preliminary and exploratory in nature. Our aim is to understand the individual contribution of each factor to epigenetic biological aging without the influence of others. Finally, since the participants in this study already have better lifestyle habits compared to the general UK population, a healthy volunteer bias may exist.

## 5. Conclusions

This study was initiated with the main objective of investigating the relationship between lifestyle patterns and epigenetic biological aging through various epigenetic clocks in British men. A significant impact of dietary and lifestyle factors on biological aging was suggested. The associations between higher intakes of saturated fat and total daily calories with accelerated biological aging, and of sufficient vitamin D intake with decelerated biological aging, as well as the negative effects of a higher BMI, underscore the critical role of nutrition and lifestyle in influencing the biological aging process. The HLI further highlights how healthier lifestyle patterns are associated with decelerated biological aging, reinforcing the potential of preventive lifestyle interventions. 

The finding that a higher HLI is associated with slower biological aging in two different biological aging clocks—PhenoAge, and DunedinPACE—will be incorporated into phase 2 of our study, the Cancer Prevention Study. The second phase of the study aims to longitudinally explore changes in biological aging before and after participation in the prevention intervention. Additionally, these results can be used in community health promotion and disease prevention programmes as an incentive.

## Figures and Tables

**Figure 1 biomedicines-12-01985-f001:**
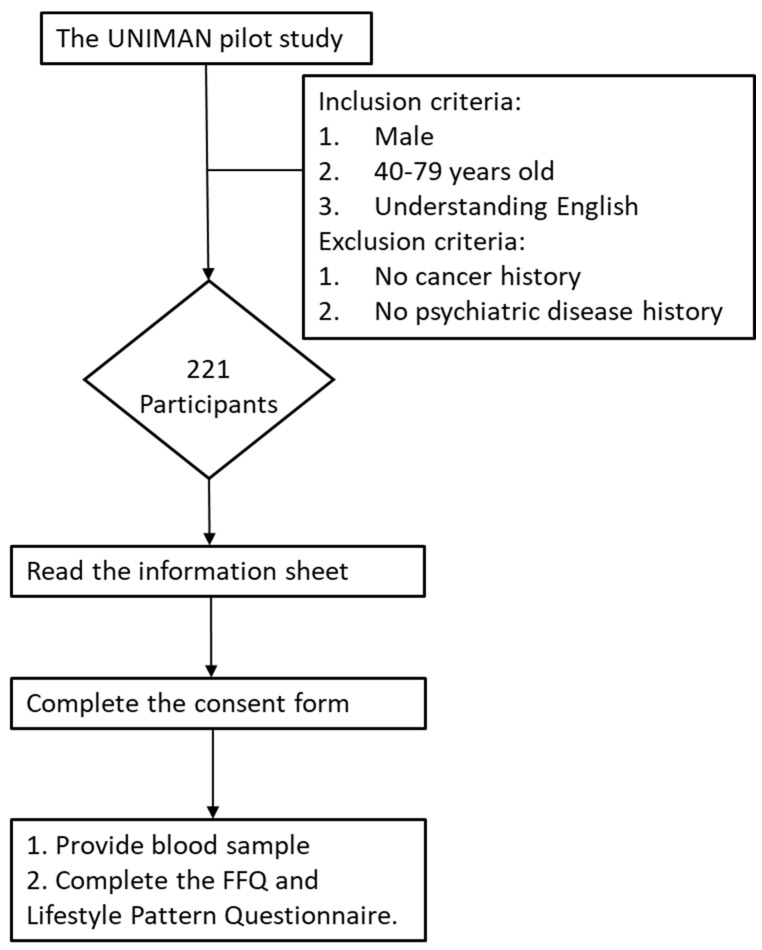
The flowchart of the recruitment of participants.

**Figure 2 biomedicines-12-01985-f002:**
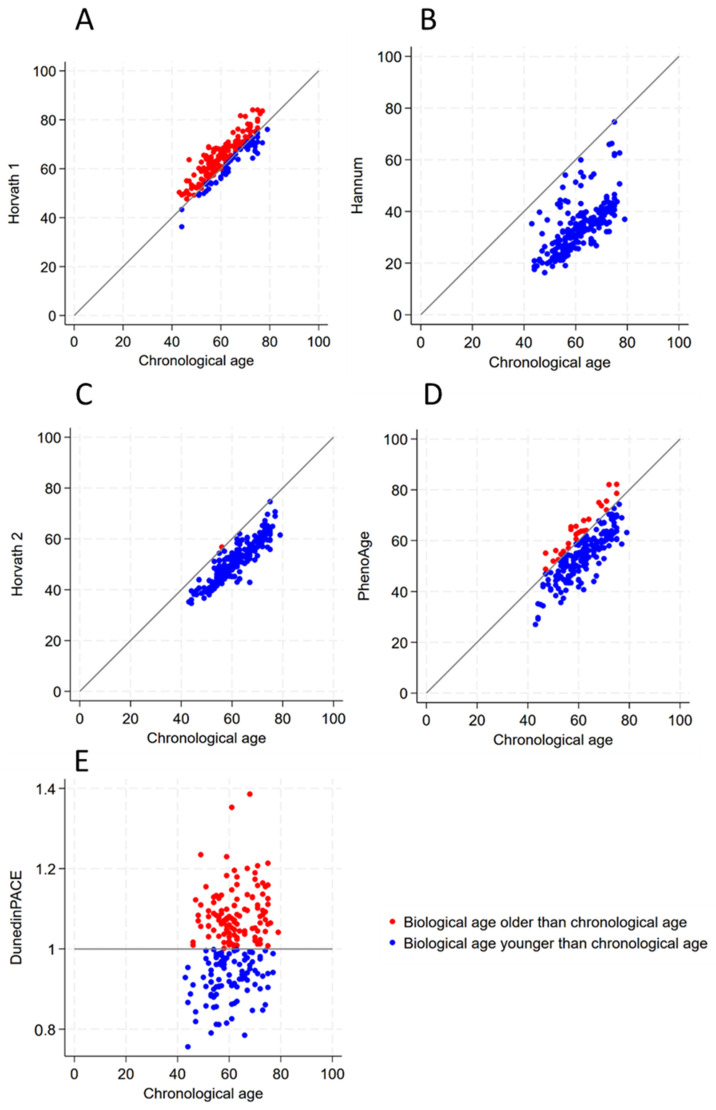
The scatter plot of different biological clock ages and chronological ages. (**A**) The comparison between Horvath 1 age and chronological age. (**B**) The comparison between Hannum age and chronological age. (**C**) The comparison between Horvath 2 age and chronological age. (**D**) The comparison between PhenoAge and chronological age. (**E**) The comparison between DunedinPACE and chronological age.

**Figure 3 biomedicines-12-01985-f003:**
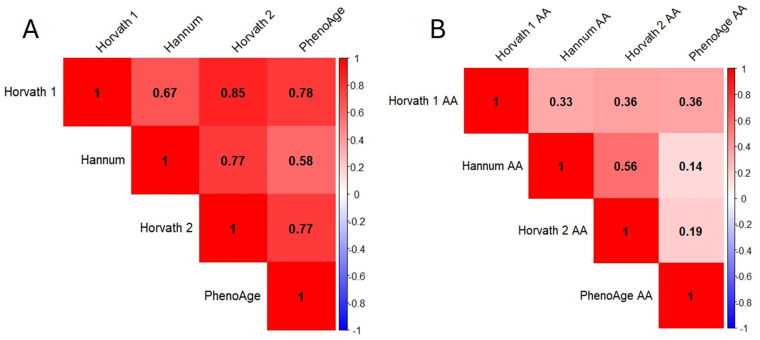
Spearman’s correlation coefficients among the different biological clock ages and the different aging acceleration (AA). (**A**) Spearman’s correlation coefficients among the different biological clock ages. All coefficients are significant at a *p*-value < 0.05. (**B**) Spearman’s correlation coefficients among the different aging acceleration (AA). All coefficients are significant at a *p*-value < 0.05.

**Table 1 biomedicines-12-01985-t001:** Information on recruitment status.

Variables	Number of Participants (n = 221)
**Recruitment**	
**A** **ge**	61.55 ± 8.27 (43–79) ^#^
**Age group**	
40–49	17 (7.69%)
50–59	77 (34.84%)
60–69	79 (35.75%)
70–79	48 (21.72%)
**Gender**	
Male	221 (100%)
**Consent to join the Lifestyle Pattern and Epigenetic Biomarker Study**	214 (96.83%)
**Food Frequency Questionnaire fill rate**	154 (69.68%)
**Lifestyle Pattern Questionnaire fill rate**	147 (66.51%)

^#^ Mean ± standard error (range).

**Table 5 biomedicines-12-01985-t005:** The Spearman’s correlation coefficients between variables and accelerated aging (AA) across different biological clocks.

	Horvath 1 AA	Hannum AA	Horvath 2 AA	PhenoAge AA	DunedinPACE
**BMI (con)**	0.067	0.012	−0.138	0.120	0.371 *
**Smoking (cat)**	−0.023	−0.234 *	−0.169	0.076	0.106
**Alcohol (cat)**	0.046	0.027	0.113	0.111	0.034
**Physical activity (cat)**	−0.050	−0.075	0.030	−0.107	0.011
**Cholesterol (con)**	0.183	0.091	0.140	0.161	−0.027
**Saturated fat (con)**	0.241 *	0.064	0.199 *	0.156	0.008
**Sugar (con)**	0.118	−0.075	0.086	0.112	−0.001
**VitD (con)**	−0.072	−0.117	0.008	−0.081	−0.199 *
**Ekal (con)**	0.221 *	−0.011	0.178	0.167	0.013

* The coefficients are significant at a false discovery rate adjusted *p*-value (*q*-value) < 0.05.

**Table 6 biomedicines-12-01985-t006:** The Pearson correlation coefficients between biological accelerated aging (AA) and the healthy lifestyle index (HLI).

	Horvath 1 AA	Hannum AA	Horvath 2 AA	PhenoAge AA	DunedinPACE
**Healthy lifestyle index (HLI)**	−0.108	0.106	0.054	−0.278 *	−0.261 *

* The coefficients are significant at a false discovery rate adjusted *p*-value (*q*-value) < 0.05.

**Table 7 biomedicines-12-01985-t007:** The univariate linear regression of healthy lifestyle index (HLI) with PhenoAge AA and DunedinPACE.

	Coefficients	SD	*p*-Value	95% CI Lower	95% CI Upper
**PhenoAge AA**	−0.780	0.224	0.001	−1.222	−0.337
**DunedinPACE**	−0.010	0.004	0.039	−0.019	−0.0005

## Data Availability

The data for this study is not publicly available due to privacy and ethical restrictions. Access to the data is limited to researchers who have received ethical review approval and consent from all the study subjects.

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
