# Peer review of "Exploring the Relationships between Lifestyle Patterns and Epigenetic Biological Age Measures in Men"

_biomedicines, 2024, doi:10.3390/biomedicines12091985_

Round 1

Reviewer 1 Report

Comments and Suggestions for Authors

This article offers valuable insights by combining five distinct epigenetic clocks—Horvath1, Hannum, Horvath2, PhenoAge, and DunedinPACE—to explore the relationship between lifestyle factors and biological aging. The comprehensive approach and robust statistical analysis present findings that are highly relevant to the epigenetic research community, particularly in understanding how lifestyle modifications can impact epigenetic aging. The manuscript is very well-written. However, a significant issue that warrants attention is the lack of clarity and detail in the inclusion and exclusion criteria for the study participants.

1. The inclusion/exclusion criteria do not specify whether metabolic disorders, such as diabetes mellitus or metabolic syndrome, were considered. Given the significant impact these conditions can have on aging, it is crucial to address this in both the main text and the inclusion/exclusion criteria. Considering that the age of participants ranged from 40-75 and the majority of them were either overweight or obese, a substantial likelihood of common metabolic disorders like type 2 diabetes mellitus, and even metabolic syndrome, is anticipated, unless addressed via rigorous screening before recruitment.

Reviewer 2 Report

Comments and Suggestions for Authors

The paper of Dr. Ke and colleagues, entitled “Exploring the relationships between lifestyle patterns and epigenetic biological age measures” represent an interesting approach, combining individual life style data with biological markers of aging. The study is multivariate, since it employs 5 types of biological age clocks’ estimators. Certainly, the paper is very interesting for a broad auditorium, from clinicians and scientists, to life-style coachers.

However, there are serious limitation, concisely reported at the manuscript. The sample size is rather modest, and the inclusion of female participants is not sufficient. The authors should either enlarge the number of women, or exclude them from the sample (and thus drive their conclusions on biological clocks in healthy Britain men). 

The authors used correlation analysis with categorical variables in the cases (such as tobacco smoking status and alcohol consumption status), which needed multivariate ANOVA (or similar methods). The authors used 5 methods to estimate the accelerated aging, but escaped the problem of corrections for multiple comparisons (Bonferroni or similar ones). It needs at least some arguments.

Reviewer 3 Report

Comments and Suggestions for Authors

The manuscript is of very high interest to readers of the scientific community.  It is also concise in its own field. I recommend acceptance and publication with some notes to the authors.

There are some minor discrepancies as it is often the case where most data originate from questionnaires and participant bias might slightly skew the outcome. Some of these are recognised and mentioned in the text, and also solution is suggested for subsequent studies (for example gender bias or relatively low response rate to dietary questionnaires). Some, however, may further be improved, in my opinion. For example smoker ratio is estimated to be low in the enrolled population based on the questionnaire. This may be checked by blood sampling (which has been performed for epigenetic studies anyway) by measuring carboxy-hemoglobin levels. Also, physical activity ratio is estimated to be high, which somewhat contradicts the average BMI of the enrolled population, but this can also be checked by wearing actigraph for a week by randomly selected volunteers.

These, however, are minor comments only and I recommend to accept the manuscript with some further conclusions for future studies. 

Round 2

Reviewer 1 Report

Comments and Suggestions for Authors

I would like to thank the authors for addressing my comments.

Author Response

Thank you so much.

Reviewer 2 Report

Comments and Suggestions for Authors

 The manuscript of Dr. Ke and colleagues, entitled “Exploring the relationships between lifestyle patterns and epigenetic biological age measures” has not been changed, but the authors wrote a rebuttal letter. I do not agree with the arguments proposed.

The sample size was moderate, and the number of female participants (less than 10%) was not sufficient to drive any type of conclusions. The authors argued that “Regarding the small proportion of female participants, we have conducted a sensitivity analysis focusing only on male participants. The results showed that the patterns of correlation between lifestyle-related factors and biological aging remained consistent.”  However, “the sensitivity analysis focusing only on male participants” does not let one to conclude anything about the female participants, nor hypothesize a presence/absence of a gender difference. So the authors should either enlarge their study, or change the declared scope of the study.

Next, the usage of correlation tests for categorical values lead to an over-estimation of the p-values (table 5). Moreover, there are only 3 active smokers (less than 2% of the sample), so the conclusion on any links between the smoking status and something measured is immature.

So, the paper is original and interesting, but the authors should present the results objectively.

Round 3

Reviewer 2 Report

Comments and Suggestions for Authors

I think that the corrections made improved the manuscript, and it can be published in the present form